# AVERAGE CERTIFIED RADIUS IS A POOR METRIC FOR RANDOMIZED SMOOTHING

## ABSTRACT

Randomized smoothing is a popular approach for providing certified robustness guarantees against adversarial attacks, and has become a very active area of research. Over the past years, the average certified radius (ACR) has emerged as the single most important metric for comparing methods and tracking progress in the field. However, in this work, we show that ACR is an exceptionally poor metric for evaluating robustness guarantees provided by randomized smoothing. We theoretically show not only that a trivial classifier can have arbitrarily large ACR, but also that ACR is much more sensitive to improvements on easy samples than on hard ones. Empirically, we confirm that existing training strategies that improve ACR reduce the model's robustness on hard samples. Further, we show that by focusing on easy samples, we can effectively replicate the increase in ACR. We develop strategies, including explicitly discarding hard samples, reweighing the dataset with certified radius, and extreme optimization for easy samples, to achieve state-of-the-art ACR, although these strategies ignore robustness for the general data distribution. Overall, our results suggest that ACR has introduced a strong undesired bias to the field, and better metrics are required to holistically evaluate randomized smoothing.

## 1 INTRODUCTION

Adversarial robustness, namely the ability of a model to resist arbitrary small perturbations to its input, is a critical property for deploying machine learning models in security-sensitive applications. Due to the incompleteness of adversarial attacks which try to construct a perturbation that manipulates the model (Athalye et al., 2018), certified defenses have been proposed to provide robustness guarantees. While deterministic certified defenses (Gowal et al., 2018; Mirman et al., 2018; Shi et al., 2021; Müller et al., 2023; Mao et al., 2023; 2024a; Palma et al., 2023; Balauca et al., 2024) incur no additional cost at inference-time, randomized certified defenses scale better with probabilistic guarantees at the cost of multiplied inference-time complexity. The most popular randomized certified defense is Randomized Smoothing (RS) (Lécuyer et al., 2019; Cohen et al., 2019), which computes the maximum certified radius for every input given the accuracy of a base model on noisy inputs $p_A$.

To train better models with larger certified radius under RS, many training strategies have been proposed (Cohen et al., 2019; Salman et al., 2019; Jeong & Shin, 2020; Zhai et al., 2020; Jeong et al., 2023). Average Certified Radius (ACR), defined to be the average of the certified radiuses over each sample in the dataset, has been the main metric to evaluate the effectiveness of these methods. However, in this work, we show that ACR is a poor metric for evaluating the true robustness of a given model under RS. We prove theoretically that ACR of a trivial classifier could be arbitrarily large given enough certification budget, and then show empirically that state-of-the-art (SOTA) RS training strategies reduce the accuracy on hard inputs to increase the ACR. Further, we demonstrate that through explicitly reweighing the training data to focus only on easy inputs, the simplest Gaussian training can be gradually amplified to achieve a SOTA ACR, questioning the development of RS training strategies.

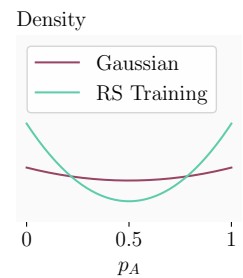

Figure 1: Conceptual illustration of the effect of RS training strategies.

**Main Contributions** Our key contributions are:

- We theoretically prove that with a large enough certification budget, ACR of a trivial classifier can be arbitrarily large, and that with the certification budget commonly used in practice, an improvement on easy inputs contributes much more to ACR than on hard inputs, more than 1000x in the extreme case (§4.1 and §4.2).

- We empirically compare RS training strategies to Gaussian training and show that all current RS training strategies are actually reducing the accuracy on hard inputs where $p_A$ is relatively small, and only focus on easy inputs where $p_A$ is very close to 1 to increase ACR (§4.3). Figure 1 conceptually visualizes this effect.

- Based on these novel insights, we develop strategies to amplify Gaussian training to achieve a SOTA ACR by reweighing the training data to focus only on easy inputs. Specifically, we discard hard inputs during training, reweigh the dataset with their contribution to ACR, and push $p_A$ extremely close to 1 for easy inputs via adversarial noise selection. With these simple modifications to Gaussian training, which do not optimize robustness for the general data distribution, but only ACR, we achieve a new SOTA in ACR (§5 and §6).

Overall, our work proves the need for new metrics for RS. In particular, we suggest to use certified accuracy at various radii as a more informative metric and encourage the community to re-evaluate existing RS training more uniformly (§7). We hope this work can inspire future research in this direction.

## 2 RELATED WORK

Randomized Smoothing (RS) is a defense against adversarial attacks that provides certified robustness guarantees (Lécuyer et al., 2019; Cohen et al., 2019). However, to achieve strong certified robustness, special training strategies tailored to RS are essential. Gaussian training, which adds Gaussian noise to the original input, is the most common strategy, as it naturally aligns with RS (Cohen et al., 2019). Salman et al. (2019) propose to add adversarial attacks to Gaussian training, and Li et al. (2019) propose a regularization to control the stability of the output. Salman et al. (2020) further shows that it is possible to exploit a pretrained non-robust classifier to achieve strong RS certified robustness with input denoising. Afterwards, Average Certified Radius (ACR), the average of RS certified radius over the dataset, is commonly used to evaluate RS training: Zhai et al. (2020) propose an attack-free mechanism that directly maximizes certified radii; Jeong & Shin (2020) propose a regularization to improve the prediction consistency; Jeong et al. (2021) propose to calibrate the confidence of smoothed classifier; Horváth et al. (2022) propose to use ensembles as the base classifier to reduce output variance; Vaishnavi et al. (2022) apply knowledge transfer on the base classifier; Jeong et al. (2023) distinguishes hard and easy inputs and apply different loss for each class. While they all improve ACR, this work shows that ACR is a poor metric for robustness, and that these training algorithms all introduce undesired side effects. This work is the first to question the development of RS training strategies evaluated with ACR and suggests new metrics as alternatives.

## 3 BACKGROUND

In this section, we briefly introduce the background required for this work.

**Adversarial Robustness** is the ability of a model to resist arbitrary small perturbations to its input. Formally, given an input set $S(x)$ and a model $f$, $f$ is adversarially robust within $S(x)$ iff for any $x_1, x_2 \in S(x)$, $f(x_1) = f(x_2)$. In this work, we focus on the $L_2$ neighborhood of an input, i.e., $S(x) := B_\epsilon(x) = \{x' \mid \|x - x'\|_2 \leq \epsilon\}$ for a given $\epsilon \geq 0$. For a given $(x, y)$ from the dataset $(\mathcal{X}, \mathcal{Y})$, $f$ is robustly correct iff $\forall x' \in S(x)$, $f(x') = y$.

**Randomized Smoothing** constructs a smooth classifier $\hat{f}(x)$ given a base classifier $f$, defined as follows: $\hat{f}(x) := \arg\max_{c \in \mathcal{Y}} \mathbb{P}_{\delta \sim \mathcal{N}(0, \sigma^2 I)}(f(x + \delta) = c)$. Intuitively, the smooth classifier assigns the label with maximum probability in the neighborhood of the input. With this formulation, Cohen et al. (2019) proves that $\hat{f}$ is adversarially robust within $B_{R(x, p_A)}(x)$ when

$p_A \geq 0.5$, where $R(x, p_A) := \sigma \Phi^{-1}(p_A)$, $\Phi$ is the cumulative distribution function of $\mathcal{N}(0, 1)$ and $p_A := \max_{c \in \mathcal{Y}} \mathbb{P}_{\delta \sim \mathcal{N}(0, \sigma^2 I)}(f(x + \delta) = c)$ is the probability of the most likely class. Average Certified Radius (ACR) is defined as the average of $R(x, p_A) I(\hat{f}(x) = y)$ over the dataset. In practice, $p_A$ cannot be computed exactly, and an estimation $\hat{p}_A$ such that $\mathbb{P}(p_A \geq \hat{p}_A) \geq 1 - \alpha$ is substituted, where $\alpha$ is the confidence threshold and $\hat{p}_A$ is computed based on $N$ trials for the event $I(f(x + \delta) = c)$. We call $N$ the certification budget, which is the number of queries to the base classifier $f$ to estimate $p_A$. Since RS certifies robustness based on the accuracy of the base model on samples perturbed by Gaussian noise, *Gaussian training*, which augments the train data with Gaussian noise, is the most common method to train the base model for RS. Specifically, it optimizes

$$\arg \min_{\theta} \mathbb{E}_{(x,y) \sim (\mathcal{X}, \mathcal{Y})} \frac{1}{m} \sum_{i=1}^{m} L(x + \delta_i; y)$$

where $\delta_i$ are sampled from $\mathcal{N}(0, \sigma^2 I)$ uniformly at random, $m$ is the number of samples and $L$ is the cross entropy loss.

## 4 WEAKNESS OF ACR AND THE CONSEQUENCE

We now first theoretically show that ACR can be arbitrarily large for a trivial classifier, assuming enough budget for certification (§4.1). We then demonstrate that with a realistic certification budget, an improvement on easy samples could contribute more than 1000 times to ACR than on hard samples (§4.2). Finally, we empirically show that due to the above weakness of ACR, all current RS training strategies evaluated with ACR reduce the accuracy on hard samples and only focus on easy samples (§4.3), improving ACR at the cost of performance on hard samples.

### 4.1 TRIVIAL CLASSIFIER WITH INFINITE ACR

In Theorem 1 below we show that for every classification problem, there exists a trivial classifier with infinite ACR given enough budget for certification, while such a classifier can only robustly classify samples from the most likely class in the dataset and always misclassifies samples from other classes.

**Theorem 1.** For every $M > 0$ and $\alpha > 0$, there exists a trivial classifier $f$ which always predicts the same class with a certification budget $N > 0$, such that the ACR of $f$ is greater than $M$ with confidence at least $1 - \alpha$.

*Proof.* Assume we are considering a $K$-class classification problem with a dataset containing $T$ samples. Let $c^*$ be the most likely class and $X^*$ be the set of all samples with label $c^*$; then there are at least $\lceil T/K \rceil$ samples in $X^*$ due to the pigeonhole theorem. We then show that a trivial classifier $f$ which always predicts $c^*$ can achieve an ACR greater than $M$ with confidence at least $1 - \alpha$ with a proper budget $N$.

Note that $p_A = 1$ for $x \in X^*$, and thus the certified radius of $x \in X^*$ is $R(x) = \sigma \Phi^{-1}(\alpha^{1/N})$. Therefore, ACR $= \frac{1}{T} \left[ \sum_{x \in X^*} R(x) + \sum_{x \notin X^*} R(x) \right] \geq \frac{1}{T} \sum_{x \in X^*} R(x) \geq \frac{1}{K} \sigma \Phi^{-1}(\alpha^{1/N})$. Setting $N = \lceil \frac{\log(\Phi(MK/\sigma))}{\log(\alpha)} \rceil + 1$, we have ACR $> M$. $\square$

Theorem 1 shows that ACR can be arbitrarily large for a trivial classifier with large enough certification budget. This implies that ACR is not reliable for evaluating a model under RS, as a trivial classifier can achieve infinite ACR with minimal robustness on at least half of classes. In practice, the budget certification $N$ is usually limited, and thus $R(x, p_A)$ is bounded by a constant for every $x$ and $p_A$. In this case, the ACR of a trivial classifier is also bounded. However, in §4.2 below, we will show that this is still problematic, as ACR is much more sensitive to improvements on easy samples than on hard samples.

### 4.2 ACR STRONGLY PREFERS EASY SAMPLES

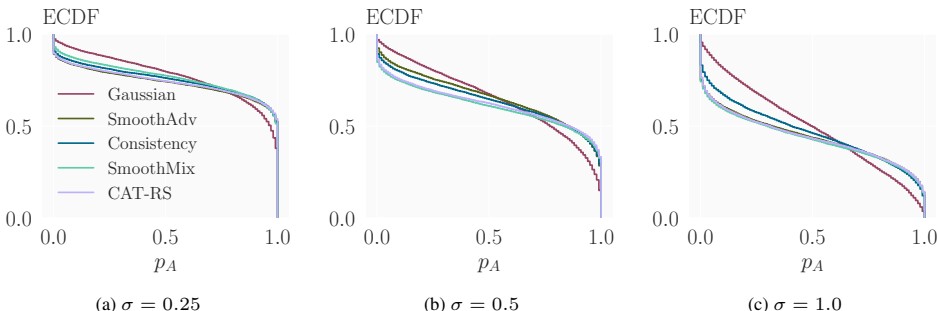

(a) $\sigma = 0.25$      (b) $\sigma = 0.5$      (c) $\sigma = 1.0$

Figure 3: The empirical cumulative distribution of $p_A$ on CIFAR-10 for models trained and certified with various $\sigma$ with different training algorithms.

We now discuss the effect of ACR with a limited budget. We follow the standard certification setting in the literature, setting $N = 10^5$ and $\alpha = 0.001$. With this budget, the maximum certified radius for one input is $R(x, p_A = 1) = \sigma\Phi^{-1}(\alpha^{1/N}) \approx 3.8\sigma$. However, we will show that $\frac{\partial R(x, p_A)}{\partial p_A}$ grows extremely fast, exceeding $1000\sigma$ when $p_A \to 1$ and close to 0 when $p_A \to 0.5$.

Without loss of generality, we set $\sigma = 1$ and denote $R(x, p_A)$ as $r$. Figure 2 shows $r$ and $\frac{\partial r}{\partial p_A}$ against $p_A$. While $r$ remains bounded by a constant 3.8, $\frac{\partial r}{\partial p_A}$

Figure 2: Certified radius $r$ and its sensitivity $\frac{\partial r}{\partial p_A}$ against $p_A$. Note the log scale of y axis in Figure 2b. $N$ is set to $10^5$, $\alpha$ is set to 0.001, and $\sigma$ is set to 1.

grows extremely fast when $p_A$ approaches 1. As a result, increasing $p_A$ from 0.99 to 0.999 improves $r$ from 2.3 to 3.0, matching the improvement achieved by increasing $p_A$ from 0 to 0.76. Therefore, to achieve maximum ACR, it is much more efficient for the training algorithm to focus on improving $p_A$ on easy samples where $p_A$ is close to 1. Further, when $p_A < 0.5$, the data point will not contribute to ACR at all, thus optimizing ACR will not increase $p_A$ with a local optimization algorithm like gradient descent. Therefore, it is natural for RS training to disregard inputs with $p_A < 0.5$ as their ultimate goal is to improve ACR.

### 4.3 RS Training Trades off Hard Samples for ACR

We have shown that ACR strongly prefers easy samples in §4.2. However, since ACR is not differentiable with respect to the model parameters because it is based on counting, RS training strategies usually do not directly apply ACR as the training loss. Instead, they optimize various surrogate objectives, and finally evaluate the model with ACR. Thus, it is unclear whether and to what extent the design of training algorithms is affected by the ACR metric. We now empirically quantify the effect, confirming the theoretical analysis. Specifically, we show that SOTA training strategies reduce $p_A$ of hard samples and put more weight (measured by gradient norm) on easy samples compared to Gaussian training.

Figure 3 shows the empirical cumulative distribution of $p_A$ for models trained with SOTA algorithms and Gaussian training. Clearly, for various $\sigma$, SOTA algorithms have higher density than Gaussian training at $p_A$ close to zero and one. While they gain more ACR due to the improvement on easy samples, hard samples are consistently underrepresented in the final model compared to Gaussian training. As a result, Gaussian training has higher $\mathbb{P}(p_A \geq 0.5)$ (clean accuracy), and SOTA algorithms exceed Gaussian training only when $p_A$ passes a certain threshold, i.e., when the certified radius is relatively large. This is problematic in practice, indicating that ACR does not properly measure the model's ability. For example, a face recognition model could have a high ACR but consistently refuse to learn some difficult faces, which is not acceptable in real-world applications.

To further quantify the relative weight between easy and hard samples indicated by each training algorithm, we measure the average gradient $l_2$ norm of easy and hard samples for models trained with different algorithms and $\sigma = 0.5$, as a proxy for the sample weight. Intuitively, samples with larger gradients contribute more to training and thus are more important for the final model. As shown in Table 1, Gaussian training puts less weight on easy samples than hard samples, which is natural as easy samples have smaller loss values. However, SOTA algorithms put more weight on

| Method | ACR | easy | hard | easy / hard |
|--------|-----|------|------|-------------|
| Gaussian | 0.56 | 10.10 | 22.67 | 0.45 |
| SmoothAdv | 0.68 | 5.60 | 5.62 | 1.00 |
| Consistency | 0.72 | 14.99 | 19.32 | 0.78 |
| SmoothMix | 0.74 | 11.72 | 11.79 | 0.99 |
| CAT-RS | 0.76 | 30.45 | 7.12 | 4.28 |

Table 1: The average gradient $l_2$ norm of easy ($p_A > 0.5$) and hard ($p_A < 0.5$) samples for models trained with different algorithms and $\sigma = 0.5$, along with their relative magnitude (*easy / hard*). The corresponding ACR is also shown.

easy samples compared to Gaussian training, e.g., the relative weight between easy and hard samples is 4.28 for CAT-RS (Jeong et al., 2023), while for Gaussian training it is 0.45. This confirms that SOTA algorithms indeed prioritize easy samples over hard samples, consistent with the theoretical analysis in §4.2.

## 5 AMPLIFYING EASY DATA GREATLY IMPROVES ACR

In §4, we concluded that ACR strongly prefers easy samples, and RS training trades off hard samples for ACR. This raises the question of whether explicitly focusing on easy data during training can effectively replicate the increase in ACR. In this section, we propose three modifications to the simplest Gaussian training to achieve this goal.

### 5.1 DISCARD HARD DATA DURING TRAINING

Samples with low $p_A$ contribute little to ACR, especially those with $p_A < 0.5$ which have no contribution at all (§4.2). Further, as shown in Figure 3, more than 20% of the data has $p_A < 0.5$ after training converges. Therefore, we propose to discard hard samples directly during training, so that they explicitly have no effect on the final training convergence. Specifically, given a warm-up epoch $E_t$ and a confidence threshold $p_t$, we discard all data samples with $p_A < p_t$ at epoch $E_t$. We fix the number of steps taken by gradient descent and re-iterate on the distilled dataset when necessary to minimize the difference in training budget. After the discard, Gaussian training also ignores hard inputs, similar to SOTA algorithms.

### 5.2 DATA REWEIGHING WITH CERTIFIED RADIUS

ACR relates non-linearly to $p_A$, and the growth of the certified radius is much faster for easy samples with high $p_A$ (Figure 2). We account for this by reweighing the data points based on their certified radius. Specifically, we use the approximate (normalized) certified radius as the weight of the probability for every data point being sampled. We formulate the weight $w$ of every data point $x$ as:

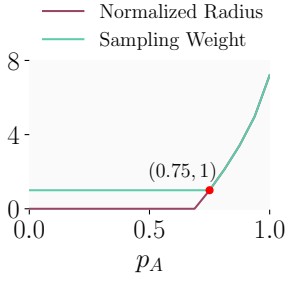

$$\hat{p}_A = \text{LOWERCONFBOUND}(C, N, 1 - \alpha)$$
$$w = \max(1, \Phi^{-1}(\hat{p}_A)/\Phi^{-1}(p_{\min})),$$

where $C$ is the count of correctly classified noisy samples and $p_{\min}$ is the reference probability threshold. Note that the estimation of $\hat{p}_A$ aligns with the certification. We normalize the weight to be at least 1 since the original radius is zero when $p_A$ is relatively low. To minimize computational overhead, we evaluate $\hat{p}_A$ every 10 epoch

Figure 4: Certified radius (divided by the value at $p_A = 0.75$) and the sampling weight of the data against $p_A$.

with $N = 16$ and $\alpha = 0.1$ throughout the paper. We set $p_{\min} = 0.75$ because this is the minimum probability that has positive certified radius under this setting. The sampling weight curve is visualized in Figure 4. After the reweighing, ACR has an approximately linear relationship to $p_A$ when $w > 1$, and easy samples are sampled more frequently than hard samples so that Gaussian training also improves $p_A$ further for easy samples.

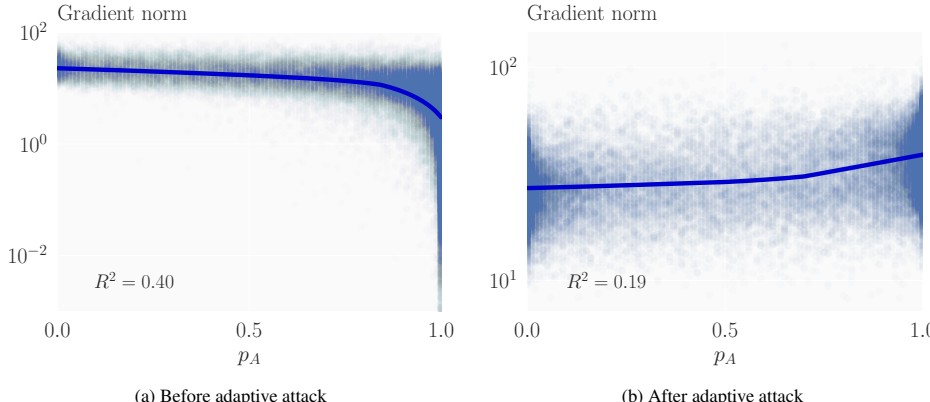

Figure 5: Comparison of the gradient norm distributions for different $p_A$ before and after the adaptive attack, $\sigma = 0.5$. Note the log scale of y axis.

## 5.3 ADAPTIVE ATTACK ON THE SPHERE

SOTA algorithms re-balance the gradient norm of easy and hard samples in contrast to Gaussian training (Table 1). In addition, when $p_A$ is close to 1, Gaussian training can hardly find a useful noise sample to improve $p_A$ further. To tackle this issue, we propose to apply adaptive attack on the noise samples to balance samples with different $p_A$. Specifically, we use Projected Gradient Descent (PGD) (Madry et al., 2018) to find the nearest noise to the Gaussian noise which can make the base classifier misclassify. Formally, we construct

$$\delta^* = \underset{f(x+\delta) \neq c, \|\delta\|_2 = \|\delta_0\|_2}{\arg\min} \|\delta - \delta_0\|_2,$$

where $\delta_0$ is a random Gaussian noise sample. Note that when $x + \delta_0$ makes the base classifier misclassify, we have $\delta^* = \delta_0$, thus hard inputs are not affected by the adaptive attack. In addition, we remark that we do not constrain $\delta^*$ to be in the neighborhood of $\delta_0$ which is adopted by CAT-RS (Jeong et al., 2023); instead, we only maintain the $l_2$ norm of the noise, thus allowing the attack to explore a much larger space. This is because for every $\delta^*$ such that $\|\delta^*\|_2 = \|\delta_0\|_2$, the probability of sampling $\delta^*$ is the same as $\delta_0$. We formalize this fact in Theorem 2.

**Theorem 2.** Assume $\delta_1, \delta_2 \in \mathbb{R}^d$ and $\delta_1 \neq \delta_2$. If $\|\delta_1\|_2 = \|\delta_2\|_2$, then $\mathbb{P}_{\mathcal{N}(0,\sigma^2 I_d)}(\delta_1) = \mathbb{P}_{\mathcal{N}(0,\sigma^2 I_d)}(\delta_2)$ for every $\sigma > 0$.

*Proof.* Let $\delta = [q_1, q_2, \ldots, q_d] \in \mathbb{R}^d$ be sampled from $\mathcal{N}(0, \sigma^2 I_d)$. Then we have

$$\mathbb{P}(\delta) = \frac{1}{(2\pi)^{d/2}\sigma^d} \exp\left(-\frac{1}{2\sigma^2} \sum_{i=1}^{d} q_i^2\right) = \frac{1}{(2\pi)^{d/2}\sigma^d} \exp\left(-\frac{1}{2\sigma^2} \|\delta\|_2^2\right).$$

This concludes the proof. $\qquad\square$

---

**Algorithm 1** Adaptive Attack

    **function** ADAPTIVEADV($f, x, c, \delta, T, \epsilon$)
    $\delta^* \leftarrow \delta$
    **for** t = 1 to $T$ **do**
      **if** $f(x + \delta^*) \neq c$ **then**
        break
      **end if**
      $\delta^* \leftarrow$ one step PGD attack on $\delta^*$ with step size $\epsilon$
      $\delta^* \leftarrow \|\delta\|_2 \cdot \delta^* / \|\delta^*\|_2$
    **end for**
    **return** $\delta^*$
    **end function**

---

Figure 5 visualizes the gradient norm distributions for different $p_A$ before and after the adaptive attack. We observe that the adaptive attack balances the gradient norm of easy and hard samples. Before the attack, the gradient norm of easy samples is much smaller than that of hard samples, while after the attack, the gradient norm of easy samples is amplified without interfering the gradient norm of hard samples. Therefore, with the adaptive attack, Gaussian training obtains a similar gradient norm distribution to SOTA algorithms, and it can find effective noise samples more efficiently. Pseudocode of the adaptive attack is shown in Algorithm 1, and more detailed description is provided in Appendix A.2.

## 5.4 OVERALL TRAINING PROCEDURE

---

**Algorithm 2** Overall Training Procedure

---

**Input:** Train dataset $\mathcal{D}$, noise level $\sigma$, hyperparameters $E_t, p_t, m, T, \epsilon$
Initialize the model $f$
**for** epoch $= 1$ to $N_{\text{epoch}}$ **do**
  **if** epoch $< E_t$ **then**
    Sample $\delta_1, \ldots, \delta_m \sim \mathcal{N}(0, \sigma^2 I)$
    Perform Gaussian training with $\delta_1, \ldots, \delta_m$
  **else**
    **if** epoch $= E_t$ **then**
      Discard hard data samples in $\mathcal{D}$ with $p_A < p_t$ to form $\mathcal{D}'$
    **end if**
    **if** epoch $\%10 = 0$ **then**
      update dataset weight according to §5.2
    **end if**
    Sample $|\mathcal{D}|$ data samples from $\mathcal{D}'$ with replacement to form the train set $\mathcal{D}''$
    **for** input $x$, label $c$ in $\mathcal{D}''$ **do**
      Sample $\delta_1, \ldots, \delta_m \sim \mathcal{N}(0, \sigma^2 I)$
      **for** i $= 1$ to $m$ **do**
        $\delta_i^* \leftarrow \text{ADAPTIVEADV}(f, x, c, \delta_i, T, \epsilon)$
      **end for**
      Perform Gaussian training with $\delta_1^*, \ldots, \delta_m^*$
    **end for**
  **end if**
**end for**

---

We now describe how the above three modifications are combined. At the beginning, we train the model with the Gaussian training (Cohen et al., 2019), which samples $m$ noisy points from the isometric Gaussian distribution uniformly at random and uses the average loss of noisy inputs as the training loss. When we reach the pre-defined warm-up epoch $E_t$, all data points with $p_A < p_t$ are discarded, and the distilled dataset is used thereafter, as described in §5.1. After this, we apply dataset reweighing and the adaptive attack to training. Specifically, every 10 epoch after $E_t$ (including $E_t$), we evaluate the model with the procedure described in §5.2 and assign the resulting sampling weight to each sample in the train set. In addition, we use the adaptive attack described in §5.3 to generate the noisy samples for training. The pseudocode is shown in Algorithm 2.

## 6 EXPERIMENTAL EVALUATION

We now evaluate our proposed method extensively. Overall, our method always achieves better ACR than SOTA methods, which indicates that focusing on easy data can effectively improve ACR.

**Baselines.** We compare our method to the following methods: Gaussian (Cohen et al., 2019), SmoothAdv (Salman et al., 2019), MACER (Zhai et al., 2020), Consistency (Jeong & Shin, 2020), SmoothMix (Jeong et al., 2021), and CAT-RS (Jeong et al., 2023). We always use the trained models provided by the authors if they are available and otherwise reproduce the results with the same setting as the original paper. We set $m = 4$ for Gaussian training and our method, since this is the standard setting for SOTA methods (Jeong et al., 2023).

| $\sigma$ | Methods | ACR | 0.00 | 0.25 | 0.50 | 0.75 | 1.00 | 1.25 | 1.50 | 1.75 | 2.00 | 2.25 | 2.50 |
|---|---|---|---|---|---|---|---|---|---|---|---|---|---|
| | Gaussian | 0.486 | **81.3** | 66.7 | 50.0 | 32.4 | 0.0 | 0.0 | 0.0 | 0.0 | 0.0 | 0.0 | 0.0 |
| | MACER | 0.529 | 78.7 | 68.3 | 55.9 | 40.8 | 0.0 | 0.0 | 0.0 | 0.0 | 0.0 | 0.0 | 0.0 |
| | SmoothAdv | 0.544 | 73.4 | 65.6 | 57.0 | 47.5 | 0.0 | 0.0 | 0.0 | 0.0 | 0.0 | 0.0 | 0.0 |
| 0.25 | Consistency | 0.547 | 75.8 | 67.4 | 57.5 | 46.0 | 0.0 | 0.0 | 0.0 | 0.0 | 0.0 | 0.0 | 0.0 |
| | SmoothMix | 0.543 | 77.1 | 67.6 | 56.8 | 45.0 | 0.0 | 0.0 | 0.0 | 0.0 | 0.0 | 0.0 | 0.0 |
| | CAT-RS | 0.562 | 76.3 | 68.1 | 58.8 | 48.2 | 0.0 | 0.0 | 0.0 | 0.0 | 0.0 | 0.0 | 0.0 |
| | **Ours** | **0.564** | 76.6 | **69.1** | **59.3** | **48.3** | 0.0 | 0.0 | 0.0 | 0.0 | 0.0 | 0.0 | 0.0 |
| | Gaussian | 0.562 | **68.7** | 57.6 | 45.7 | 34.0 | 23.7 | 15.9 | 9.4 | 4.8 | 0.0 | 0.0 | 0.0 |
| | MACER | 0.680 | 64.7 | 57.4 | 49.5 | 42.1 | 34.0 | 26.4 | 19.2 | 12.0 | 0.0 | 0.0 | 0.0 |
| | SmoothAdv | 0.684 | 65.3 | **57.8** | 49.9 | 41.7 | 33.7 | 26.0 | 19.5 | 12.9 | 0.0 | 0.0 | 0.0 |
| 0.5 | Consistency | 0.716 | 64.1 | 57.6 | 50.3 | 42.9 | 35.9 | 29.1 | 22.6 | 16.0 | 0.0 | 0.0 | 0.0 |
| | SmoothMix | 0.738 | 60.6 | 55.2 | 49.3 | 43.3 | 37.6 | 32.1 | 26.4 | 20.5 | 0.0 | 0.0 | 0.0 |
| | CAT-RS | 0.757 | 62.3 | 56.8 | 50.5 | **44.6** | 38.5 | 32.7 | 27.1 | 20.6 | 0.0 | 0.0 | 0.0 |
| | **Ours** | **0.760** | 59.3 | 54.8 | 49.6 | 44.4 | **38.9** | **34.1** | **29.0** | **23.0** | 0.0 | 0.0 | 0.0 |
| | Gaussian | 0.534 | **51.5** | **44.1** | 36.5 | 29.4 | 23.8 | 18.2 | 13.1 | 9.2 | 6.0 | 3.8 | 2.3 |
| | MACER | 0.760 | 39.5 | 36.9 | 34.6 | 31.7 | 28.9 | 26.4 | 23.8 | 21.1 | 18.6 | 16.0 | 13.8 |
| | SmoothAdv | 0.790 | 43.7 | 40.3 | 36.9 | 33.8 | 30.5 | 27.0 | 24.0 | 21.4 | 18.4 | 15.9 | 13.4 |
| 1.0 | Consistency | 0.757 | 45.7 | 42.0 | **37.8** | 33.7 | 30.0 | 26.3 | 22.9 | 19.6 | 16.6 | 13.9 | 11.6 |
| | SmoothMix | 0.788 | 42.4 | 39.4 | 36.7 | 33.4 | 30.0 | 26.8 | 23.9 | 20.8 | 18.6 | 15.9 | 13.6 |
| | CAT-RS | 0.815 | 43.2 | 40.2 | 37.2 | **34.3** | 31.0 | 28.1 | 24.9 | 22.0 | 19.3 | 16.8 | 14.2 |
| | **Ours** | **0.844** | 42.0 | 39.4 | 36.5 | 33.9 | **31.1** | **28.4** | **25.6** | **23.1** | **20.6** | **18.3** | **16.1** |

Table 2: Comparison of certified test accuracy (%) at different radii and ACR on CIFAR-10. The best and the second best results are highlighted in bold and underline, respectively; for certified accuracy, we highlight those that are worse than Gaussian training at the same radius in gray.

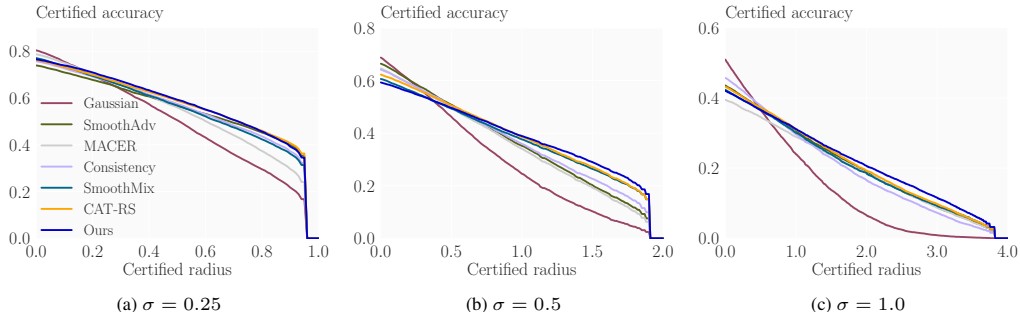

(a) $\sigma = 0.25$        (b) $\sigma = 0.5$        (c) $\sigma = 1.0$

Figure 6: Certified radius-accuracy curve on CIFAR-10 for different methods.

**Main Result.** Table 2 shows the ACR of different methods on CIFAR-10. Detailed description of the training, including applied hyperparameters, is provided in Appendix A.1. Our method consistently outperforms all baselines on ACR, which confirms the effectiveness of our modification to Gaussian training in increasing ACR. Further, our method successfully increases the certified accuracy at large radius, as we explicitly focus on easy inputs which is implicitly taken by other SOTA methods. Figure 6 further visualizes certified accuracy at different radii, showing that at small radii, SOTA methods (including ours) are systematically worse than Gaussian training, while at large radii, these methods consistently outperform Gaussian training. The success of our simple and intuitive modification to Gaussian training suggests that ACR introduces a systematic bias in method selection, and that the field should re-evaluate RS training strategies with better metrics.

**Ablation Study.** We present a thorough ablation study in Table 3. When applied alone, all three components of our method improve ACR compared to Gaussian training. Combing two components arbitrarily improves the ACR compared to using only one component, and the best ACR is achieved when all three components are combined. This confirms that each component contributes to the improvement of ACR. In addition, they mostly improve the certified accuracy at large radii and reduce certified accuracy at small radii, which is consistent with our intuition that focusing on easy inputs can improve the ACR. More ablation on the hyperparameters is provided in Appendix B.

| $\sigma$ | discard | dataset weight | adversarial | ACR | 0.00 | 0.25 | 0.50 | 0.75 | 1.00 | 1.25 | 1.50 | 1.75 | 2.00 | 2.25 | 2.50 |
|---|---|---|---|---|---|---|---|---|---|---|---|---|---|---|---|
| | | Gaussian | | 0.486 | 81.3 | 66.7 | 50.0 | 32.4 | 0.0 | 0.0 | 0.0 | 0.0 | 0.0 | 0.0 | 0.0 |
| | ✔ | | | 0.515 | 81.2 | 69.3 | 53.7 | 36.8 | 0.0 | 0.0 | 0.0 | 0.0 | 0.0 | 0.0 | 0.0 |
| | | ✔ | | 0.512 | 81.3 | 69.4 | 53.3 | 36.3 | 0.0 | 0.0 | 0.0 | 0.0 | 0.0 | 0.0 | 0.0 |
| 0.25 | | | ✔ | 0.537 | 76.7 | 66.7 | 55.6 | 44.3 | 0.0 | 0.0 | 0.0 | 0.0 | 0.0 | 0.0 | 0.0 |
| | ✔ | ✔ | | 0.523 | 81.1 | 69.7 | 54.6 | 38.3 | 0.0 | 0.0 | 0.0 | 0.0 | 0.0 | 0.0 | 0.0 |
| | ✔ | | ✔ | 0.550 | 77.4 | 68.5 | 57.7 | 45.4 | 0.0 | 0.0 | 0.0 | 0.0 | 0.0 | 0.0 | 0.0 |
| | | ✔ | ✔ | 0.554 | 75.0 | 67.1 | 58.1 | 48.1 | 0.0 | 0.0 | 0.0 | 0.0 | 0.0 | 0.0 | 0.0 |
| | ✔ | ✔ | ✔ | **0.564** | 76.6 | 69.1 | 59.3 | 48.3 | 0.0 | 0.0 | 0.0 | 0.0 | 0.0 | 0.0 | 0.0 |
| | | Gaussian | | 0.525 | 65.7 | 54.9 | 42.8 | 32.5 | 22.0 | 14.1 | 8.3 | 3.9 | 0.0 | 0.0 | 0.0 |
| | ✔ | | | 0.627 | 68.4 | 59.4 | 49.5 | 39.4 | 29.0 | 20.5 | 13.0 | 7.0 | 0.0 | 0.0 | 0.0 |
| | | ✔ | | 0.662 | 68.1 | 59.7 | 50.3 | 41.1 | 31.7 | 23.7 | 16.2 | 9.2 | 0.0 | 0.0 | 0.0 |
| 0.5 | | | ✔ | 0.701 | 63.4 | 56.2 | 49.1 | 41.7 | 34.5 | 28.2 | 22.1 | 16.5 | 0.0 | 0.0 | 0.0 |
| | ✔ | ✔ | | 0.672 | 68.5 | 60.1 | 51.0 | 41.8 | 32.4 | 24.2 | 16.7 | 9.3 | 0.0 | 0.0 | 0.0 |
| | ✔ | | ✔ | 0.731 | 63.4 | 56.8 | 50.1 | 43.7 | 37.0 | 30.8 | 24.4 | 18.0 | 0.0 | 0.0 | 0.0 |
| | | ✔ | ✔ | 0.741 | 56.1 | 52.1 | 47.3 | 43.2 | 38.6 | 34.1 | 29.1 | 23.1 | 0.0 | 0.0 | 0.0 |
| | ✔ | ✔ | ✔ | **0.760** | 59.3 | 54.8 | 49.6 | 44.4 | 38.9 | 34.1 | 29.0 | 23.0 | 0.0 | 0.0 | 0.0 |
| | | Gaussian | | 0.534 | 51.5 | 44.1 | 36.5 | 29.4 | 23.8 | 18.2 | 13.1 | 9.2 | 6.0 | 3.8 | 2.3 |
| | ✔ | | | 0.665 | 46.8 | 42.1 | 37.6 | 33.1 | 28.7 | 24.3 | 20.2 | 16.1 | 12.6 | 9.8 | 7.4 |
| | | ✔ | | 0.695 | 49.9 | 44.9 | 39.7 | 34.9 | 29.8 | 25.3 | 21.4 | 17.5 | 13.6 | 10.1 | 7.1 |
| 1.0 | | | ✔ | 0.690 | 47.3 | 42.0 | 37.0 | 32.0 | 27.1 | 23.2 | 19.9 | 16.6 | 13.6 | 10.8 | 8.4 |
| | ✔ | ✔ | | 0.736 | 48.8 | 44.5 | 40.3 | 35.9 | 31.4 | 27.3 | 22.9 | 19.1 | 15.2 | 11.7 | 8.7 |
| | ✔ | | ✔ | 0.771 | 47.0 | 43.1 | 39.1 | 35.0 | 30.9 | 27.1 | 23.6 | 20.2 | 17.0 | 14.1 | 11.6 |
| | | ✔ | ✔ | 0.818 | 39.7 | 37.5 | 34.8 | 32.5 | 29.9 | 27.7 | 25.3 | 22.6 | 20.1 | 17.9 | 15.8 |
| | ✔ | ✔ | ✔ | **0.844** | 42.0 | 39.4 | 36.5 | 33.9 | 31.1 | 28.4 | 25.6 | 23.1 | 20.6 | 18.3 | 16.1 |

Table 3: Ablation study on each component in our method.

# 7 DISCUSSION

We have shown that ACR does not uniformly represent the robustness of a model for different data. Therefore, the field has to seek better alternative metrics to evaluate the robustness of models under RS. We suggest to use certified accuracy at various radii as a more informative metric, which can be easily computed with the same certification budget as ACR. In addition, since achieving maximum certified accuracy at all radii is a challenging task, we should allow algorithms to customize models for different radii, including the certification hyperparameter $\sigma$. This is similar to the practice in deterministic certified training (Gowal et al., 2018; Mirman et al., 2018; Shi et al., 2021; Müller et al., 2023; Mao et al., 2023; 2024a;b; Palma et al., 2023; Balauca et al., 2024).

While our modifications presented in §5 are not designed to improve robustness for the general data distribution, they show effectiveness in increasing certified accuracy at large radius. Other existing algorithms show similar effects. Therefore, it is important to note that the field has indeed made progress over the years. However, new metrics which considers robustness more uniformly should be developed to evaluate RS, and algorithms that outperform generally at various radii are encouraged. We hope this work can inspire future research in this direction.

# 8 CONCLUSION

This work rigorously demonstrates that Average Certified Radius (ACR) is a poor metric for Randomized Smoothing (RS). Theoretically, we prove that ACR of a trivial classifier can be arbitrarily large, and that an improvement on easy inputs contributes much more to ACR than on hard inputs. Empirically, we show that all state-of-the-art (SOTA) strategies reduce the accuracy on hard inputs and only focus on easy inputs to increase ACR. Based on these novel insights, we develop strategies to amplify Gaussian training by reweighing the training data to focus only on easy inputs. Specifically, we discard hard inputs during training, weight the dataset with their contribution to ACR, and apply extreme optimization for easy inputs via adversarial noise selection. With these intuitive modifications to the simple Gaussian training, we replicate the effect of SOTA training algorithms and achieve a new SOTA ACR. Overall, our results suggest the need for evaluating RS training with better metrics.

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

| $\sigma$ | 0.25 | 0.5 | 1.0 |
|---|---|---|---|
| $E_t$ | 60 | 70 | 60 |
| $p_t$ | 0.5 | 0.4 | 0.4 |
| $T$ | 3 | 6 | 4 |
| $\epsilon$ | 0.25 | 0.25 | 0.5 |

Table 4: Hyperparameters we use on CIFAR-10.

## A EXPERIMENT DETAILS

### A.1 TRAINING DETAILS

We follow the standard training protocols in previous works. Specifically, we use ResNet-110 (He et al., 2016) for CIFAR-10. We investigate three noise levels, $\sigma = 0.25, 0.5, 1.0$. We train the model for 150 epochs with an initial learning rate of 0.1, which is decreased by a factor of 10 in every 50 epochs. Stochastic gradient descent (SGD) optimizer with a momentum of 0.9 is used, and the batch size is set to 256. We adjust the following hyperparameters to find the best performance for each setting:

$$E_t = \text{the epoch to discard hard data samples}$$
$$p_t = \text{the threshold of } p_A \text{ to discard hard data samples}$$
$$T = \text{the maximum number of steps of the PGD attack}$$
$$\epsilon = \text{the step size of the PGD attack}$$

All other parameters are fixed to their default values. Specifically, we use 100 noise samples ($N = 100$) to calculate $p_A$ when discarding hard samples. For updating dataset weight, we use 16 noise samples ($N = 16$) to calculate $p_A$, with a minimum value set to 0.75 ($p_{\min}$=0.75). The number of noise samples for each input is set to $m = 4$. The hyperparameters we use on CIFAR-10 are shown in Table 4.

We use the CERTIFY function in Cohen et al. (2019) to calculate the certified radius, same as previous baselines, where $N = 10^5$ and $\alpha = 0.001$.

### A.2 ADVERSARIAL ATTACK ALGORITHM

In the adaptive attack, we implement the $l_2$ PGD and $l_\infty$ PGD attack. The difference between them lies in how the noise is updated based on the gradient. For $l_2$ PGD, the noise is updated as follows:

$$\delta^* = \delta + \epsilon \cdot \nabla_\delta L(f(x + \delta), y) / \|\nabla_\delta L(f(x + \delta), y)\|_2,$$

while for $l_\infty$ PGD, the noise is updated as follows:

$$\delta^* = \delta + \epsilon \cdot \text{sign}(\nabla_\delta L(f(x + \delta), y)),$$

where $L$ is the loss function, $f$ is the model, $x$ is the input, $y$ is the label, $\delta$ is the noise, $\delta^*$ is the updated noise, sign is the sign function, and $\epsilon$ is the step size. Without specific instructions, our experiments are always conducted with $l_2$ PGD. We show the results with $l_\infty$ PGD on CIFAR-10 in Appendix C.2.

## B ADDITIONAL ABLATION STUDIES

In this section, we provide additional ablation studies on the effect of different hyperparameters. All results are based on $\sigma = 0.5$ and CIFAR-10. Unless otherwise specified, all hyperparameters are the same as those in Table 4.

### B.1 EFFECT OF DISCARDING HARD DATA

We investigate the effect of the two hyperparameters $E_t$ and $p_t$ in discarding hard samples. The results are shown in Table 5 and Table 6. If hard samples are discarded too early, it leads to a

| $E_t$ | ACR | 0.00 | 0.25 | 0.50 | 0.75 | 1.00 | 1.25 | 1.50 | 1.75 |
|---|---|---|---|---|---|---|---|---|---|
| 50 | 0.738 | 58.0 | 53.2 | 48.3 | 42.9 | 38.1 | 33.0 | 27.5 | 21.8 |
| 70 | 0.760 | 59.3 | 54.8 | 49.6 | 44.4 | 38.9 | 34.1 | 29.0 | 23.0 |
| 90 | 0.751 | 60.1 | 54.9 | 49.6 | 44.1 | 38.6 | 33.4 | 27.5 | 21.4 |
| 110 | 0.746 | 60.0 | 55.1 | 49.5 | 43.7 | 38.0 | 33.3 | 27.2 | 21.0 |
| 130 | 0.738 | 61.4 | 56.1 | 50.2 | 44.2 | 37.5 | 31.7 | 25.5 | 19.2 |

Table 5: Effect of the discarding epoch $E_t$.

| $p_t$ | ACR | 0.00 | 0.25 | 0.50 | 0.75 | 1.00 | 1.25 | 1.50 | 1.75 |
|---|---|---|---|---|---|---|---|---|---|
| 0.3 | 0.753 | 59.5 | 54.5 | 49.4 | 44.3 | 39.0 | 33.7 | 28.0 | 21.9 |
| 0.4 | 0.760 | 59.3 | 54.8 | 49.6 | 44.4 | 38.9 | 34.1 | 29.0 | 23.0 |
| 0.5 | 0.757 | 60.0 | 54.7 | 49.9 | 44.5 | 38.9 | 33.7 | 27.9 | 22.1 |
| 0.6 | 0.758 | 59.2 | 54.6 | 49.5 | 44.3 | 39.4 | 34.1 | 28.7 | 22.2 |
| 0.7 | 0.749 | 59.5 | 54.4 | 49.0 | 43.7 | 38.7 | 33.2 | 27.7 | 21.7 |
| 0.8 | 0.740 | 59.5 | 54.4 | 49.1 | 43.6 | 38.1 | 32.8 | 26.8 | 20.8 |

Table 6: Effect of the discarding $p_A$ threshold $p_t$.

decrease in accuracy for all radii, as many potential easy samples are discarded before their $p_A$ reaches $p_t$. Conversely, discarding hard samples too late allows more samples to remain, resulting in improved clean accuracy but reduced performance on easy samples, as the training focuses less on them.

### B.2 EFFECT OF ADAPTIVE ATTACK

For the adaptive attack, we evaluate the effect of the step size $\epsilon$ and the number of steps $T$. The results are shown in Table 7 and Table 8. In general, increasing the number of steps strengthens the attack, which helps find effective noise samples for extremely easy inputs. As a result, performance improves at larger radii, but relatively less attention is given to hard samples, leading to a decrease in clean accuracy. Similarly, increasing the step size in a moderate range ($\epsilon < 1.0$) has a similar effect. However, if the step size is too large, the attack loses effectiveness, resulting in decreased performance at all radii.

## C ADDITIONAL RESULTS

### C.1 RATIO OF REMAINED DATA AFTER DISCARDING HARD SAMPLES

The remaining data ratios after discarding hard samples under different $\sigma$ on CIFAR-10 are 91%, 80% and 58%, when $\sigma = 0.25$, 0.5 and 1.0 respectively. The results indicate that even though a significant portion of the training set is discarded during training, we still achieve a better ACR than the current SOTA model according to Table 2. This indicates that discarding a large amount of data during training leads to a better ACR, proving once again that ACR may not accurately represent the overall robustness of the model.

| $\epsilon$ | ACR | 0.00 | 0.25 | 0.50 | 0.75 | 1.00 | 1.25 | 1.50 | 1.75 |
|---|---|---|---|---|---|---|---|---|---|
| 0.125 | 0.744 | 64.2 | 58.1 | 51.6 | 44.7 | 38.1 | 31.4 | 24.4 | 17.1 |
| 0.250 | 0.760 | 59.3 | 54.8 | 49.6 | 44.4 | 38.9 | 34.1 | 29.0 | 23.0 |
| 0.500 | 0.703 | 50.8 | 47.4 | 43.9 | 40.3 | 36.4 | 32.9 | 28.8 | 24.4 |
| 1.000 | 0.624 | 43.2 | 40.7 | 38.2 | 35.4 | 32.7 | 29.7 | 26.6 | 22.7 |
| 2.000 | 0.533 | 38.1 | 35.6 | 33.0 | 30.5 | 27.6 | 24.9 | 22.0 | 19.0 |

Table 7: Effect of the adversarial attack step size $\epsilon$.

| $T$ | ACR | 0.00 | 0.25 | 0.50 | 0.75 | 1.00 | 1.25 | 1.50 | 1.75 |
|---|---|---|---|---|---|---|---|---|---|
| 2 | 0.719 | 66.2 | 58.9 | 51.5 | 43.7 | 36.2 | 28.9 | 21.8 | 14.5 |
| 4 | 0.752 | 62.0 | 56.7 | 50.9 | 44.6 | 38.6 | 32.7 | 26.3 | 19.7 |
| 6 | 0.760 | 59.3 | 54.8 | 49.6 | 44.4 | 38.9 | 34.1 | 29.0 | 23.0 |
| 8 | 0.748 | 57.0 | 52.6 | 48.0 | 43.4 | 38.7 | 34.2 | 29.2 | 23.6 |
| 10 | 0.732 | 54.1 | 49.9 | 45.9 | 42.0 | 37.8 | 34.0 | 29.5 | 24.6 |

Table 8: Effect of the number of steps of adversarial attack $T$.

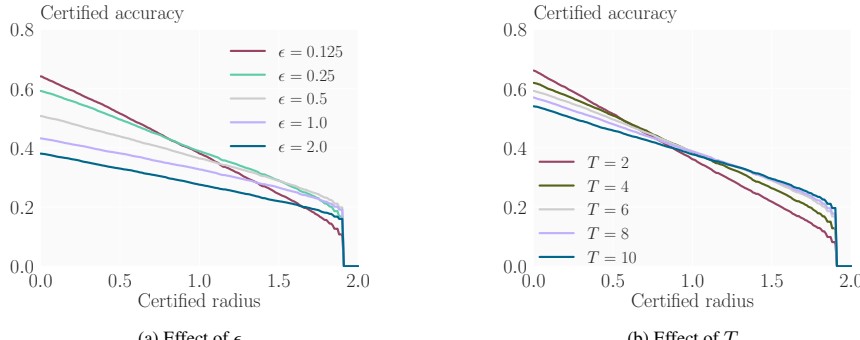

(a) Effect of $\epsilon$        (b) Effect of $T$

Figure 7: Certified radius-accuracy curves on CIFAR-10 for different $\epsilon$ and $T$.

| $\sqrt{d} \cdot \epsilon$ | ACR | 0.00 | 0.25 | 0.50 | 0.75 | 1.00 | 1.25 | 1.50 | 1.75 |
|---|---|---|---|---|---|---|---|---|---|
| 0.125 | 0.722 | 66.3 | 59.6 | 51.8 | 44.3 | 36.4 | 29.1 | 21.3 | 13.7 |
| 0.250 | 0.746 | 62.7 | 57.0 | 51.1 | 44.8 | 38.4 | 32.0 | 25.4 | 18.5 |
| 0.500 | 0.727 | 55.7 | 51.3 | 46.9 | 42.5 | 37.7 | 32.8 | 28.0 | 22.6 |
| 1.000 | 0.640 | 46.4 | 43.3 | 39.9 | 36.7 | 33.5 | 30.0 | 25.9 | 22.0 |

Table 9: ACR and certified accuracy under the $l_\infty$ PGD attack with $\sigma = 0.5$ on CIFAR-10 with different step size.

| $T$ | ACR | 0.00 | 0.25 | 0.50 | 0.75 | 1.00 | 1.25 | 1.50 | 1.75 |
|---|---|---|---|---|---|---|---|---|---|
| 3 | 0.718 | 66.0 | 59.0 | 51.6 | 43.8 | 36.2 | 28.8 | 21.2 | 14.0 |
| 4 | 0.731 | 65.0 | 58.5 | 51.5 | 44.2 | 37.3 | 29.9 | 23.1 | 15.8 |
| 5 | 0.736 | 64.2 | 57.8 | 51.4 | 44.6 | 37.7 | 30.6 | 23.7 | 16.5 |
| 6 | 0.746 | 62.7 | 57.0 | 51.1 | 44.8 | 38.4 | 32.0 | 25.4 | 18.5 |
| 7 | 0.743 | 61.5 | 56.4 | 50.6 | 44.4 | 38.1 | 32.4 | 25.7 | 19.1 |
| 8 | 0.742 | 60.1 | 54.8 | 49.7 | 43.7 | 38.0 | 32.7 | 26.9 | 20.3 |

Table 10: ACR and certified accuracy under the $l_\infty$ PGD attack with $\sigma = 0.5$ on CIFAR-10 with different number of steps.

## C.2 EXPERIMENTS ON $l_\infty$ PGD ATTACK

In this section, we show the results under $l_\infty$ PGD attack in Table 9 and Table 10. For $l_\infty$ attack, we use much smaller step sizes $\epsilon$ to have the similar attack strength to $l_2$ attack. Specifically, we investigate $\sqrt{d} \cdot \epsilon = 0.125, 0.25, 1.5, 1.0$ for $l_\infty$ attack, where $d$ is the dimension of the input. After the thorough experiments, we find that using $\sqrt{d} \cdot \epsilon = 0.25$ and the same with $l_2$ attack for other hyperparameters reaches the best ACR.

The the $l_\infty$ attack has the similar effect on results to the $l_2$ attack. Increasing the step size or the number of steps of the attack will result in better accuracy at large radii but lower accuracy at small radii.

