# OpenReview forum: "Average Certified Radius is a Poor Metric for Randomized Smoothing"
_ICLR.cc/2025/Conference — Submitted to ICLR 2025_

### Official Review · Reviewer_Xqxu · 2024-10-21

**Soundness:** 3
**Presentation:** 2
**Contribution:** 3
**Rating:** 6
**Confidence:** 5

**Summary:**

This paper evaluates the effectiveness of **Average Certified Radius (ACR)**, a popular metric for assessing model robustness against adversarial attacks, particularly within the context of **Randomized Smoothing (RS)**. The authors specifically target **existing training strategies**, showing that these methods provide a misleading sense of general robustness.

They prove that a trivial classifier can achieve arbitrarily high ACR with a large certification budget, despite being unreliable. Additionally, they demonstrate that ACR disproportionately benefits from improvements on "easy" samples, with gains potentially over 1000x greater than those on "hard" samples. Through experiments, they reveal that state-of-the-art (SOTA) RS training methods inflate ACR by focusing on easy inputs while sacrificing robustness on harder samples. To further critique ACR, the authors propose strategies to boost ACR artificially, such as discarding hard inputs and reweighing data, reinforcing the bias toward easy samples.

The paper concludes by highlighting the limitations of ACR, presenting experimental evidence, and advocating for a shift in how the field evaluates robustness in **randomized smoothing**, suggesting **certified accuracy at various radii** as a more reliable alternative metric.

**Strengths:**

1. **Originality and Significance:** The paper presents an original and critical perspective on a widely accepted metric in adversarial robustness, the **Average Certified Radius (ACR)**, which has been largely unchallenged in the literature. By demonstrating that ACR is a flawed metric for evaluating randomized smoothing (RS) models, the authors provide a fresh lens through which the community can reevaluate robustness. By exposing the weaknesses of ACR, the paper has significant implications for future research in the field of certified defenses.

2. **Quality:** The theoretical analysis is well-supported by empirical results, making the paper technically sound. The authors back their theoretical claims with well defined experiments that verify their theoretical findings. The experiments are well-designed, with appropriate baselines and metrics to validate the claims. The paper covers a lot of **training based RS stratagies** to support their claims which further adds to the experimental aspect of this work.

**Weaknesses:**

1. **Limited Scope**: The paper lacks proper clarity on one aspect, wether it's findings are specific to **training-based RS methods** or **Randomized Smoothing** in general.  If I understood correctly, than the theoretical analysis in section 4.1 is not specific to **training based RS methods**, but during evaluation the paper mainly critiques **ACR** for **training-based RS methods** but doesn't extend/discusses the impact of theoretical analysis to **other RS methods**, which also rely on ACR to evaluate certified robustness. Expanding the critique to cover these methods would provide a more comprehensive view and increase the impact of the findings.

2. **Lack of Practical Guidance**: While the authors propose **certified accuracy at various radii** as an alternative metric, they don’t provide detailed guidance on how to implement it effectively in practice. Offering concrete steps or case studies on how this metric can be integrated into evaluation pipelines would make the solution more actionable. Furthermore, **certified accuracy at various radii* is a metric which is more commonly used, compared to ACR, to evaluate **other RS methods**, hence this further limits the scope of the work as it adds to the point that this work is only relevant to **training-based RS methods**.

**Questions:**

I mainly have two questions based on the points mentioned in weaknesses above, If they can be answered properly, specifically question 1, then I am willing to increase the rating.

1. Could you clarify how the limitations of **ACR** apply to **other RS methods**? Since ACR is also used in these methods to evaluate certified robustness, how do your findings extend to these approaches? It would be helpful to understand whether ACR is equally flawed in these contexts. Or if the work is specific to **training-based RS methods**?

2. You propose using **certified accuracy at various radii** as an alternative to ACR. Could you provide more specific guidance on how this should be applied in practice? For example, how should practitioners choose appropriate radii, and how would this approach balance performance across easy and hard samples?

---

> ### Author Response · Authors · 2024-11-19
> **Response to $\Rx$**
>
> We are happy to hear that $\Rx$ considers our work technically sound, our experiments well-designed, our critique to the metric ACR original and critical, and our results having significant implications for future research. In the following, we address all concrete questions raised by Reviewer $\Rx$.
>
> **Q1: General RS (certification) also uses ACR to evaluate (certification methods). Could the findings derived for RS training be extended (to RS certification) as well? Is ACR equally flawed (in RS certification)?**
>
> While the problems of ACR extend to certification settings, this is not problematic in certification. This is because a fixed model is usually considered by certification, and thus a certification algorithm needs to achieve universal improvements on certified accuracy at various radii. As a result, ACR is usually not considered by certification algorithms. For example, the most recent SOTA certification work [1] only evaluates with certified accuracy at various radii and does not use ACR. Therefore, while ACR is equally flawed, in RS certification it is not a problem because it is not widely used. In contrast, ACR is commonly used to evaluate RS training algorithms, particularly when claiming SOTA performance.
>
> **Q2: You propose certified accuracy at various radii as an alternative to ACR. Could you provide more specific guidance on how this should be applied in practice? For example, how should practitioners choose appropriate radii, and how would this approach balance performance across easy and hard samples?**
>
> Great question! We provide a preliminary discussion in Section 7 and are happy to include more specific instructions. Regarding the choice of appropriate radii, we show in Table 2 that there is only one separation, namely small and large certified radius. Therefore, while asking for numbers at all radius currently considered by the field (from $0$ to $2.5$ with a step of $0.25$) are possible, we believe using two settings representing small and large certified radius, respectively, is enough to cover the important aspects of the algorithms. For example, one may set the certified radius to be 0.25 and 1.5, and ask for clean and certified accuracy of models at these two radii. Further, as discussed in Section 7, different model selection, i.e., different sets of hyperparameters should be allowed for each setting. This aligns with the practice in the closely related area of deterministic certified training. By fixing the certified radius, easy samples cannot contribute more than hard samples, thus resolving the evaluation tradeoff.
>
>
> *Reference*
>
> [1] Lyu, Saiyue, Shadab Shaikh, Frederick Shpilevskiy, Evan Shelhamer, and Mathias Lécuyer. Adaptive Randomized Smoothing: Certifying Multi-Step Defences against Adversarial Examples, 2024.

---

### Official Review · Reviewer_BinL · 2024-10-22

**Soundness:** 3
**Presentation:** 3
**Contribution:** 2
**Rating:** 5
**Confidence:** 4

**Summary:**

The paper discusses the limitations of using the Average Certified Radius (ACR) as a primary metric for evaluating robustness in Randomized Smoothing (RS), a technique used to defend machine learning models against adversarial attacks. The authors argue that ACR disproportionately focuses on easy samples and can be artificially inflated, even by trivial classifiers. Their key contributions include:
1.	Theoretical Analysis: The paper shows that a trivial classifier can achieve arbitrarily large ACR given a sufficient certification budget, proving that ACR is unreliable as a robustness measure.
2.	Empirical Findings: They demonstrate that existing state-of-the-art (SOTA) RS training strategies, which improve ACR, reduce model accuracy on hard samples, thus prioritizing easy samples.
3.	Proposed Solutions: The authors introduce training modifications—such as discarding hard samples, reweighing data based on certified radius, and applying adaptive adversarial attacks—that achieve a new SOTA ACR by focusing only on easy samples. These strategies improve ACR but ignore general robustness.

**Strengths:**

This paper presents a fact that the Average Certified Radius (ACR) is an unfair metric for evaluating robustness in Randomized Smoothing (RS), which is helpful to future research on Randomized Smoothing.

**Weaknesses:**

1. Novelty is relatively limited. The first contribution suggests that the Average Certified Radius (ACR) can be increased arbitrarily by enlarging the sample size. However, [1] also finds that certified accuracy is closely related to sample size. Figure 4(a) in [1] demonstrates that as pA -> 1.0, the certified radius significantly increases with sample size. The existence of [1] diminishes the novelty of this paper, which also fails to reference this closely related work.

2. The logic of this paper is somewhat unnatural. While it proves that ACR is a poor metric because it favors easy samples, it would be more logical to design a more reasonable metric. Instead, the paper proposes a new training algorithm that gives more weight to easy samples, exploiting the weakness of the ACR metric to increase ACR.

3. The proposed training algorithm requires using Projected Gradient Descent (PGD) to compute adversarial examples, which is computationally expensive. This is why methods like MACER and consistency, choose attack-free training algorithms to mitigate computation costs.

[1] Input-Specific Robustness Certification for Randomized Smoothing. AAAI 2022.

**Questions:**

1. Recognizing that ACR is an inadequate metric, please include a more detailed discussion on designing a more effective comparison metric.
2. The training algorithm presented is based on adversarial methods; please add a comparison regarding computational overhead.

---

> ### Author Response · Authors · 2024-11-19
> **Response to $\Rb$**
>
> We are happy to hear that Reviewer $\Rb$ agrees that our work reveals ACR as a poor metric for randomized smoothing and benefits future research. In the following, we address all concrete questions raised by Reviewer $\Rb$.
>
> **Q1: Previous work [1] has shown (empirically) that ACR correlates with the sample size and the effect is more pronounced when $p_A$ is close to 1. Does this limit the contribution of this work?**
>
> Certified radius under randomized smoothing is known to correlate with sample size [1,2]. However, to the best of our knowledge, we are first to formally present that it is unbounded for a trivial classifier. This has a different consequence, as the former is a reasonable and acceptable property, while the latter proves high ACR does not imply good robustness. Therefore, we consider this a novel and more impactful property of ACR.
>
> We present that ACR is much more sensitive to $p_A$ when $p_A$ is close to 1. We note that [1] (Figure 4.a) presents that the benefit of their proposed input-specific sampling over input-agnostic sampling increases when $p_A$ is close to 1, which is not directly related to our result. This can also be seen from the decrease of the benefit when $p_A \in [0.5, 0.95]$, shown in Figure 4.a in [1], while our work (Figure 2.b) shows a consistent incline in sensitivity. These facts suggest that their result is not comparable to ours, as we have different trends and study different subjects. Therefore, we consider this novel as well.
>
> **Q2: This work finds ACR favors easy samples and proposes a new algorithm exploiting this to increase ACR. Is this logic unnatural?**
>
> We would like to highlight that the proposed algorithm serves a different purpose. Instead of establishing a new algorithm that works better than previous works, it aims to further solidify our argument about the badness and consequences of using ACR for evaluation. We note that the proposed algorithm consists of three independent tricks, all exploiting the bad properties of ACR rather than improving robustness for the general data distribution. Despite high ignorance of robustness, this algorithm is able to beat existing algorithms in ACR. Therefore, this proves that an algorithm with high ACR does not necessarily have high robustness, which further supports our claim that ACR should no longer be used to evaluate RS training algorithms. This logic has been made clear in the abstract, the introduction and the technical sections, but we will highlight it further to avoid future confusion.
>
> **Q3: PGD is used in the proposed algorithm, which increases the computational cost. Could you provide a comparison regarding the computational overhead?**
>
> Based on the discussion above, the proposed algorithm is not for real use, thus we do not consider the increase in cost to be a problem. However, to quantify the exact computational overhead, we provide a comparison of model training time used in Table 2 below. Costs are all measured on a single NVIDIA RTX 4090 GPU.
>
>
> |        Method        | Epochs | Training time (hrs) |
> |:--------------------:|:------:|:-------------------:|
> |        CAT-RS        |    150  |         6.1         |
> |        SmoothMix   |    150  |         3.4         |
> |      Consistency     |    150  |         0.8         |
> |       SmoothAdv   |     150   |        17.0    |
> |         MACER        |    440  |         18.8        |
> |       Gaussian       |   150  |         1.4         |
> | Ours ($\sigma=0.25$) |    150  |         3.0         |
> |  Ours ($\sigma=0.5$) |   150  |         4.0         |
> |  Ours ($\sigma=1.0$) |   150  |         3.6         |
>
>
> **Q4: Recognizing that ACR is a poor metric, could you discuss and design a more effective comparison metric?**
>
> We discuss alternative metrics in Section 7. Certified accuracy at various radii is suggested for replacement. We note that this is not equivalent to the currently applied *certified accuracy at various radii for a single model* metric. In contrast, we suggest following the practice in the highly related field of deterministic certified training, namely allowing different model selections given a predefined certified radius. For example, one may choose one set of hyperparameters to train a model performing well at radius 0.5, and another set of hyperparameters to train another model performing well at radius 1. This unifies the contribution of easy and hard samples to the metric. We will make this more clear in the revised manuscript.
>
> *Reference*
>
> [1] Ruoxin Chen, Jie Li, Junchi Yan, Ping Li, Bin Sheng. Input-Specific Robustness Certification for Randomized Smoothing, 2022.
>
> [2] Jeremy M. Cohen, Elan Rosenfeld, and J. Zico Kolter. Certified adversarial robustness via randomized smoothing, 2019.

---

### Official Review · Reviewer_SA4J · 2024-11-07

**Soundness:** 3
**Presentation:** 3
**Contribution:** 1
**Rating:** 3
**Confidence:** 5

**Summary:**

This submission studies the metric of average certified radius (ACR) in randomized smoothing training literature. First, the submission theoretically shows that ACR is ill-posed in terms of unboundedness for trivial toy classifiers and uneven sensitivity for easy and hard samples. Then, harnessing these features, a training method that overly focuses on easy samples can hack the metric to achieve a state-of-the-art ACR.

**Strengths:**

1. Revealing the problem of ACR to the randomized smoothing community, which rectifies the past error and promotes a more scientific evaluation protocol.

2. Technically sound and clear writing.

**Weaknesses:**

1. The impact is limited. ACR is a rather specific metric that mainly exists in the randomized smoothing literature. Moreover, in the literature, ACR is used in conjunction with certified accuracy under various radii - as a poor metric, the community is not overly focused on the metric yet.

2. Technical contribution is limited. It is relatively obvious to see the unboundedness of ACR (for a constant classifier) and the sensitivity with respect to $P_A$ is a direct consequence of $\Phi^{-1}(\cdot)$'s bounded domain and unbounded value domain.

3. The paper is overly focused on a narrow topic and can be benefited from better contextualization. For example, for randomized smoothing, diffusion-purification is an active and developing method, such as "(Certified!!) Adversarial Robustness for Free!", DensePure, and DiffSmooth; ensemble-based training is also widely studied such as Horvath et al (boosting) and Yang et al (on the certified...).


Minor:
The implication of Figure 1 on page 1 is not clear.

**Questions:**

No specific questions.

---

> ### Author Response · Authors · 2024-11-19
> **Response to $\Rs$**
>
> We are happy to hear that Reviewer $\Rs$ considers our work technically sound, written clearly, and revealing that the de-facto evaluation metric for randomized smoothing, ACR, has severe problems. In the following, we address all concrete questions raised by Reviewer $\Rs$.
>
> **Q1: Certified accuracy under various radii is provided in literature in conjunction with ACR; is the community overly focused on ACR?**
>
> Indeed, certified radius under various radii for a single selected model is provided together with ACR in the literature, e.g., Table 1 in the most recent SOTA work CAT-RS [1] and Table 2 in our work. However, we note that such numbers are only for completeness and are not the main reference of SOTA claims. For example, as shown in Table 2 in this work as well as Table 1 in [1], the claimed SOTA algorithms do not achieve the best certified accuracy under various radii uniformly, especially at small radius. Instead, achieving the best ACR is the main (and usually the only) support for claiming SOTA. The same observation extends to most works in this area, as can be seen in Table 2 in this work. Therefore, we believe that it is reasonable to conclude that the community almost relies solely on the ACR metric to compare algorithms, especially when claiming SOTA performance.
>
> **Q2: The unboundedness of ACR and the sensitivity with respect to $p_A$ is relatively easy to reveal; does this limit the contribution of this paper?**
>
> We agree that the proof of the unboundness of ACR and its sensitivity is not super hard. However, an intuitive and easy-to-understand proof does not limit the impact of such results. To the best of our knowledge, this is the first time that such facts are formally presented. In addition, despite these bad properties of ACR being seemingly easy to reveal, the community still relies heavily on ACR to evaluate RS training algorithms, as discussed above. Without this work, the community would potentially continue to use this poor metric, ignoring its problems. Therefore, we believe this solidly established result, that ACR is a poor metric, is of significant impact to the field.
>
> **Q3: The paper doesn’t investigate other active and developing randomized smoothing methods, e.g., synthetic data and ensemble-based training. Is it overly focused on a narrow topic (that ACR is a poor metric for randomized smoothing), and thus the impact is limited?**
>
> This paper focuses on developing an important conclusion: ACR is a poor metric for randomized smoothing, regardless of the evaluated algorithm. We consider this seemingly narrow topic to be sufficiently impactful, thus devote the full work to the solid establishment of this conclusion. We first prove the bad theoretical properties of ACR, and then reveal how exploiting these bad properties is enough to develop a new “SOTA” algorithm that only wins in ACR. As discussed in the last question, without this conclusion, the community would continue to use the wrong metric, leading to bad consequences, e.g., developing new SOTA (in ACR) algorithms that do not improve robustness. Therefore, we believe this result is of significant interest to the field.
>
> Given the studied subject, we mainly focus on evaluating and comparing to more algorithmic works, namely those that only change the training algorithm rather than training data and the model. Therefore, using synthetic data and better models is to be considered orthogonal to this work. While covering more comparisons does not harm, it also does not help to further solidify our conclusion: ACR is poor. We believe that this conclusion extends to any study that evaluates and claims SOTA based on ACR solely.
>
> **Q4: What is the implication of Figure 1?**
>
> Figure 1 is a conceptual illustration of Figure 3. This extracts the core patterns shown in Figure 3, showing that advanced RS training strategies lead to more low-confidence and high-confidence samples ($p_A$ close to 0 and 1) than Gaussian training. It tries to convey an easy-to-understand message in the introduction, without overwhelming details for new readers.
>
> *Reference*
>
> [1] Jongheon Jeong, Seojin Kim, and Jinwoo Shin. Confidence-aware training of smoothed classifiers for certified robustness, 2024.

---

> ### Comment · Reviewer_SA4J · 2024-11-26
>
> Thanks for the response.
>
> For Q1, I respectfully disagree with the argument. CAT-RS is published at AAAI 2023, while “(Certified!!) Adversarial Robustness for Free!” is at ICLR 2023 - later than CAT-RS and *never* use the problematic ACR to claim state-of-the-art performance. Follow-up work for the ICLR 2023 includes DensePure and Diffsmooth - both are published, well-received by the community, and *never* use ACR as the metric. The ACR supports the SOTA's claim for a very limited number of papers that are not the latest (less than 5, IMHO).
>
> For Q2, the community does not rely heavily on ACR. I agree that it is important to point out the flaw in ACR. But given the limited scope, it may be more suitable to discuss in a dedicated workshop.
>
> For Q3, same as Q1 and Q2 - only a special group of training for randomized smoothing works rely on ACR. If broader approaches for improving certified robustness are considered orthogonal to this work - then it further justifies the limited scope of this work.
>
> Thanks for explaining Q4; maybe the meaning can be added to the caption of Figure 1.
>
> Given the clarified facts, I am more confident on my review, and I increased the confidence score accordingly.

---

> ### Author Response · Authors · 2024-11-26
> **Response to $\Rs$ follow-up**
>
> We are glad that all technical concerns from Reviewer $\Rs$ have been cleared. We regret to hear that the reviewer has strong opinions about the scope of ACR, thus we further clarify it below.
>
> Regarding the time of formal publication, we think it is better to compare the time of the release of the mentioned works, i.e., denoising-based approaches [1,2] vs CAT-RS [3]. This is because some works are quickly accepted while others are slower, thus comparing their formal publication time does not indicate the time of their work. Therefore, it is more reasonable to compare the time when they are publicly known. According to their arXiv record, both [1] and [2] are released earlier than CAT-RS [3]. Therefore, this definitely suggests that algorithmic works such as CAT-RS are still evaluated based on ACR, and the community still willingly accepts it (as indicated by its formal publication).
>
> Further, we believe some denoising-based works [1,2] do not evaluate with ACR for a reason different to what we have shown. They are based on the seminal work [4] released in 2020, which does not evaluate with ACR but instead with certified accuracy at various radii. This is also because they use off-the-shelf classifiers, while algorithmic works such as CAT-RS trains custom classifiers. Nevertheless, ACR is popular after the early work [4], which definitely suggests that the community does not realize the problems of ACR. Therefore, while denoising-based works [1,2] directly compare with [4] and thus do not use ACR to evaluate, we think this is not because they explicitly realize the problems of ACR; rather, this is because they want to be consistent and comparable with the most related work [4]. This is also supported by the fact that [1,2] do not mention a single problem of ACR as shown in this work.
>
> In addition, the most recent denoising-based works, including DiffSmooth [5] and [6] (both published), also evaluate with ACR. [5] and [6] are released roughly one year after [1] and [2], and they evaluate with ACR in almost all main tables (Table 3, 4 and 5 in [5]; Table 1, 2, 3 and 5 in [6]). This further supports that even denoising-based methods are not aware of the problems of ACR, and the deviation of [1] and [2] from evaluating with ACR is simply due to consistency with prior works. Therefore, we would like to highlight that existing literature evidently suggests ACR is extensively used, even in the most recent denoising-based methods.
>
> We hope this further discussion clears Reviewer $\Rs$'s doubt about the scope of ACR in the community, and welcome further questions.
>
> **Reference**
>
> [1] (Certified!!) Adversarial Robustness for Free!, https://arxiv.org/abs/2206.10550
>
> [2] DensePure: Understanding Diffusion Models towards Adversarial Robustness, https://arxiv.org/abs/2211.00322
>
> [3] Confidence-aware Training of Smoothed Classifiers for Certified Robustness, https://arxiv.org/abs/2212.09000
>
> [4] Denoised Smoothing: A Provable Defense for Pretrained Classifiers, https://arxiv.org/abs/2003.01908
>
> [5] DiffSmooth: Certifiably Robust Learning via Diffusion Models and Local Smoothing, https://arxiv.org/abs/2308.14333
>
> [6] Multi-scale Diffusion Denoised Smoothing, https://arxiv.org/abs/2310.16779

---

> > ### Comment · Reviewer_SA4J · 2024-11-26
> >
> > Thanks for the clarification.
> >
> > First, [1,2,4,5] do not rely on ACR to justify their main contributions. Even though [5] has ACR results, it is mainly for completeness and is not demonstrated in the main results in Tables 1 and 2 and the abstract. Second, I think it is better to compare the time of peer-reviewed publication. I understand the disagreement, and we will leave it open. Third, the publications heavily relying on ACR for result demonstration mostly come from a few research groups and are far from a community consensus. Besides, the general randomized smoothing works beyond training do not heavily rely on ACR. It falls out of common practice not to even mention these related works, putting the manuscript detached from the literature and showing a biased landscape of the research state of the art.
> >
> > The research value is appreciated, and I believe it would be a good publication in a dedicated workshop (with research groups that used ACR heavily attending).

---

> ### Author Response · Authors · 2024-11-27
> **Follow-up-2 $\Rs$**
>
> We thank Reviewer $\Rs$ for the reply and acknowledgement of our contributions. It seems that both ours and the reviewer’s opinions have been made clear, and there are no factual concerns left. We would like to further highlight that it is made clear in our manuscript that we discuss ACR in the context of training algorithms, and such algorithms relying on ACR for evaluation, as shown in our related work section and the discussions with Reviewer $\Rs$.

---

### Author Response · Authors · 2024-11-19
**General Response**

$\newcommand{\Rs}{\textcolor{green}{SA4J}}$
$\newcommand{\Rb}{\textcolor{blue}{BinL}}$
$\newcommand{\Rx}{\textcolor{purple}{Xqxu}}$


We thank all reviewers for their insightful reviews, helpful feedback, and interesting questions. We are particularly encouraged that all reviewers agree that we solidly establish the claim we make in the title: ACR is a poor metric for randomized smoothing. Further, our work is considered technically sound ($\Rs$, $\Rx$), written clearly ($\Rs$) and important to future work ($\Rx$). No shared concern from reviewers is identified, thus we will address all concrete questions in individual responses.

---

### Meta-Review · Area_Chair_oAKi · 2024-12-13

**Metareview:**

This paper is the first to show that  the average certified radius (ACR) is a poor metric for evaluating robustness provided by randomized smoothing. The idea is interesting. But reviewers think the technical contribution is limited and topics of this paper is too narrow. It is better to appear in the workshop of the specific conferences.

**Additional Comments On Reviewer Discussion:**

reviewers think the technical contribution is limited and topics of this paper is too narrow.

---

### Decision · Program_Chairs · 2025-01-22

Reject